



**Semi-Annual Variation of Excited Hydroxyl Emission at Mid-Latitudes**
Mykhaylo Grygalashvyly[1], Alexander I. Pogoreltsev[2], Alexey B., Andreyev[3], Sergei P.
Smyshlyaev[2], and Gerd R. Sonnemann[1]
[1]Leibniz-Institute of Atmospheric Physics at the University Rostock in Kühlungsborn,
Schloss-Str.6, D-18225 Ostseebad Kühlungsborn, Germany
[2]Department of Meteorological Forecasting, Russian State Hydrometeorological University
(RSHU), Saint-Petersburg, Russia
[3]Institute of the Ionosphere, Almaty, Kazakhstan
**Abstract**
Ground-based observations show a phase shift in semi-annual variation of excited hydroxyl
(OH*) emissions at mid-latitudes (43° N) compared to those at low latitudes. This differs
from the annual cycle at high latitudes. We examine this behaviour utilising an OH* airglow
model which was incorporated into the 3D chemistry-transport model (CTM). Through this
modelling, we study the morphology of the excited hydroxyl emission layer at mid-latitudes
(30° N -50° N), and we assess the impact of the main drivers of its semi-annual variation:
temperature, atomic oxygen, and air density. We found that this shift in the semi-annual cycle
is determined mainly by the superposition of annual variations of temperature and atomic
oxygen concentration. Hence, the winter peak for emission is determined exclusively by
atomic oxygen concentration, whereas the summer peak is the superposition of all impacts,
with temperature taking a leading role.
**1. Introduction**
Since the second half of the 20th century, emissions of excited hydroxyl have been
used for three main purposes: 1) to infer information about temperature and its long-term



change; 2) to obtain distributions of minor chemical constituents ($O_3$, H, and O) at the
altitudes of the mesopause; and 3) to investigate dynamic processes such as tides, gravity, and
planetary waves (GWs and PWs, respectively), sudden stratospheric warmings (SSWs),  and
quasi-biennial oscillation (QBO).
Hence, a number of authors have studied temperatures in the mesopause region using airglow
emission ground-based observations focusing on long-term trends (e.g., Bittner et al., 2002;
Holmen et al., 2014; Dalin et al., 2020, and references therein) with attention to seasonal
variations (e.g., Reid et al., 2017, and references therein) and the solar-cycle effect (e.g.,
Kalicinsky et al., 2016, and references therein).
Minor chemical constituents as well as chemical heat have also been retrieved by OH*
emission observations. Ever since atomic oxygen concentration was determined by the rocket-
born detection of OH* airglow (Good, 1976), this method has come into wide use for
obtaining information about distributions of minor chemical constituents in the mesopause
region, namely, atomic oxygen concentration (e.g., Russell et al., 2005; Mlynczak et al.,
2013a, and references therein), ozone concentration (e.g., Smith et al., 2009, and references
therein), atomic hydrogen concentration (e.g., Mlynczak et al., 2014, and references therein),
and exothermic chemical heat (e.g., Mlynczak et al., 2013b, and references therein).
Numerous works using airglow observations, have been devoted to dynamic processes, for
example, to study mesopause variabilities in time of SSWs (Damiani et al., 2010; Shepherd et
al., 2010). Gao et al. (2011) studied the temporal evolution of nightglow brightness and height
during SSW events. A year earlier, they found a QBO signal in the excited hydroxyl emission
(Gao et al., 2010). The climatology of PWs was investigated in works by Takahashi et al.
(1999), Buriti et al. (2005), and Reisin et al. (2014). Tides were studied by Xu et al. (2010)
and Lopez-Gonzalez et al. (2005). GW parameters based on the airglow technique were
investigated, for example, by Taylor et al. (1991) and Wachter et al. (2015). A more complete



description of works in which hydroxyl emissions were used to study dynamic processes can
be found in a review by Shepherd et al. (2012).
The morphology of the OH* layer is an essential component in the interpretation of
observations and in understanding the processes involved in layer variability. Annual
variations in the OH* layer have been identified at all latitudes (Marsh et al., 2006).
Equatorial and low-latitude semi-annual variations have been observed by satellites (e.g.,
Abreu and Yee, 1989; Liu et al., 2008, and references therein), as well as by ground-based
instruments (Takahashi et al., 1995), and they have been modelled by several research teams
(Le Texier et al., 1987; Marsh et al., 2006, and references therein). The maxima of emissions
were found to occur near equinoxes. In spite of the large number of studies on this subject,
there are still knowledge gaps. Recently, unexpected behaviour in the semi-annual cycle of
excited hydroxyl emission has been found by ground-based observations, with a shift of the
peaks from equinoxes to summer and winter at middle latitudes (Popov et al., 2018; Popov et
al., 2020); this was also found by modelling (Grygalashvyly et al., 2014, Fig. 3). Similar
variations in OH* emissions with peaks near equinoxes have been observed at middle
latitudes (34.6° N) in the southern hemisphere (Reid et al., 2014). These results were provided
without explanations; in our short paper, we offer a preliminary explanation.
The second chapter of our manuscript describes the observational technique and model that
were applied; in the third chapter, we present results and an analysis of observations and
modelling; conclusions are provided in the fourth chapter.

**2. Observational technique and model**

**2.1. Observational technique**



The spectral airglow temperature imager (SATI), which measures nightglow intensity for
vibrational transitions of $OH^*_{v=6} \rightarrow OH^*_{v=2}$ and temperature using vibrational-rotational
transitions, was assembled at the Institute of Ionosphere (43° N, 77° E) in Almaty,
Kazakhstan. It represents a Fabry-Perot spectrometer with a CCD (charge-coupled device)
camera as a detector and a narrow-band interference filter as the etalon. Following Lopez-
Gonzalez et al. (2007), we use an interference filter with the centre at 836.813 nm and a
bandwidth of 0.182 nm. This corresponds to the spectral region of the OH*(6-2) band. In
order to infer the temperature, the calculated spectra for different vibro-rotational transitions
are compared with those from observations. The SATI operates at a sixty-second exposure
that provides corresponding time resolution. The method of temperature retrieval is well-
described by Lopez-Gonzalez et al. (2004). The observations of temperature were validated
using satellite SABER measurements (Lopez-Gonzalez et al., 2007; Pertsev et al., 2013).
Additional details about this instrument are presented in many papers (Wies et al., 1997;
Aushev et al., 2000; Lopez-Gonzalez et al., 2004, 2005, 2007, 2009).

**2.2. Model and numerical experiment**

The model of excited hydroxyl (MEH) calculates the OH* number densities at each
vibrational level $v$ as the production divided by losses (excited hydroxyl is assumed in the
photochemical equilibrium), which include the chemical sources as well as collisional and
emissive removal:

$$
\quad [OH_v] = \frac{\left( \begin{array}{c} \varsigma_v a_1 [O_3][H] + \psi_v a_2 [O][HO_2] + \sum_{v'=v+1}^{9} B_{v'v}[O_2][OH_{v'}] + C_{v+1}[N_2][OH_{v+1}] + \\ + \sum_{v'=v+1}^{9} D_{v'v}[O][OH_{v'}] + \sum_{v'=v+1}^{9} E_{v'v}[OH_{v'}] \end{array} \right)}{\left( \begin{array}{c} a_3(v)[O] + \sum_{v''=0}^{v-1} D_{vv''}[O] + C_v[N_2] + \\ + \sum_{v''=0}^{v-1} B_{vv''}[O_2] + \sum_{v''=0}^{v-1} E_{vv''} \end{array} \right)}, \left( \begin{array}{c} v < v' \\ v'' < v \end{array} \right). \quad (1)
$$

The first term in the numerator of (1) is the reaction $O_3 + H \rightarrow OH_v + O$, where $a_1$ is the
reaction rate, and $\varsigma_v$ represents the branching ratios (Adler-Golden, 1997). The second term is





the $O + HO_2 \rightarrow OH_v + O_2$ reaction, where $a_2$ and $\psi_v$ are the reaction rate and nascent
distribution, respectively (Kaye (1988) after Takahashi and Batista (1981)). The other three
summands represent the populations resulting from collisional relaxation from higher $v$-
levels, where $B$, $C$, and $D$ are the collisional deactivation coefficients for $O_2$ (Adler-Golden,
1997), $N_2$ (Makhlouf et al., 1995), and O (Caridade et al., 2013), respectively. The last
summand is the multi-quantum population by spontaneous emissions, where $E_{v'v}$ is the
spontaneous emission coefficient (Xu et al., 2012). The losses occur, additionally, through the
chemical removal of the excited hydroxyl by atomic oxygen, where $a_3(v)$ is the vibrationally
dependent reaction rate (Varandas et al., 2004). The calculations in Eq. (1) are incorporated
into the chemistry-transport model (CTM).
Here, we enumerate only the main features of the CTM as one can find extended descriptions
in manyworks (Sonnemann and Grygalashvyly, 2020; Grygalashvyly et al., 2014; and
references therein). The CTM consists of four blocks: chemical, transport, radiative, and
diffusive. The chemical block accounts for 19 constituents, and 63 photo-dissociations and
chemical reactions (Burkholder et al., 2015). The chemical code utilises a family approach
with the odd-oxygen (O($^1$D), O, $O_3$), odd-hydrogen (H, OH, $HO_2$, $H_2O_2$), and odd-nitrogen
(N($^2$D), N($^4$S), NO, $NO_2$) families (Shimazaki, 1985). In the radiative part, the dissociation
rates are taken from a pre-calculated table depending on zenith angle and altitude (Kremp et
al., 1999). The transport block calculates advections in three directions following Walcek
(2000). The diffusive part accounts for only vertical molecular plus turbulent diffusion
(Morton and Mayers, 1994). This model has been validated against observations of ozone,
which plays a role in the formation of OH* (e.g., Hartogh et al., 2011; Sonnemann et al.,
2007; and references therein) and water vapour, which is the principal source of odd-
hydrogens and, particularly, of atomic hydrogen (e.g., Hartogh et al., 2010; Sonnemann et al.,
2008; and references therein). Our current analysis used the run for year 2009, which was





published and described in a number of works (Grygalashvyly et al., 2014, section 4;
Sonnemann et al., 2015).
Here we assume that the structures in the longitudinal direction are equivalent to local time
(LT) behaviour, with 24 LT related to midnight at 0° longitude. The LTs of successive
longitudes are used to analyse our calculations. Hence, in the following figures related to the
model results, longitude is used as the so-called 'pseudo time'. The night-time averaged
values account for the period from 21:45 LT to 2:15 LT. For the purposes of our discussion,
we use 'pressure-altitude' (or other words 'pseudo-altitude') $Z^* = -H \ln(P/P_0)$, where $P$
represents pressure: $P_0 = 1013 \ mbar$ is the surface pressure, and $H = 7 \ km$ is the scale
height.

**3. Results and discussion**

Figure 1a illustrates the nightly mean monthly averaged values of the observed annual
variability of intensity at 43° N (red line) and the modelled annual variability of volume
emission at the peak of the OH* layer at 43.75° N (black line), both for transition
$OH^*_{v=6} \rightarrow OH^*_{v=2}$. The error bar shows standard deviation. By the observations as well as by
modelling, we can clearly see semi-annual variations of emissions with peaks in winter and
summer.
Grygalashvyly et al. (2014), Sonnemann et al. (2015), and Grygalashvyly (2015) have derived
and confirmed through modelling that the concentration of excited hydroxyl (hence, volume
emission and intensity) at peak is directly proportional to the product of the surrounding
pressure (hence, it depends on altitude), atomic oxygen number density, and the negative
power of temperature (Eq. A2 in the Appendix). Thus, in order to infer the reasons for this
semi-annual variation, one should consider three drivers of OH* variability: temperature,
atomic oxygen concentration, and height of the layer.





Figure 1b shows the monthly mean nightly averaged values of the observed annual variability
of temperature at 43° N (red line) and the modelled annual variability of temperature at the
$OH^*_{v=6}$ peak at 43.75° N (black line). Both the observations and the modelling show minima
in summer and maxima in winter. Hence, the temperature decline can be one of the reasons
for the summer intensity (and volume emission) peak.
Figures 1c and 1d depict modelled monthly mean nightly averaged values of atomic oxygen at
$OH^*_{v=6}$ peak and the height of the excited hydroxyl peak, respectively, at 43.75° N. The
modelling shows the peaks of atomic oxygen concentration in July and December–January,
with the largest values in winter. The variation of height through the year occurs from ~90 km
to 94 km. This is an essential variability and provides input to the variability of the
concentration of the surrounding air.
In order to study the morphology of this semi-annual variation and assess the impacts of
temperature, atomic oxygen concentration, and height (concentration of air) variability, we
calculate one-month sliding averaged values based on the model results. Figure 2 illustrates
the modelled annual variability at the $OH^*_{v=6}$ peak: a) volume emission ($OH^*_{v=6} \rightarrow OH^*_{v=2}$), b)
temperature, c) atomic oxygen concentration, and d) height.
The summer maximum of volume emission (Fig. 2a) shows the strongest values in July and is
extended from ~30° N to ~50° N. The summer maximum is stronger than that in winter. The
winter maximum has its strongest values in January and a positive gradient into the winter
pole direction; at latitudes 30°–50° N, it represents the rest of the annual variation at high
latitudes that occurs because of the annual variation in general mean circulation and fluxes of
atomic oxygen which correspond to this variability (Liu et al., 2008; Marsh et al., 2006).
Similar behaviour of the emissions for transition $OH^*_{v=8} \rightarrow OH^*_{v=3}$ was captured by WINDII
(Wind Imaging Interferometer) and modelled by Thermosphere-Ionosphere-Mesosphere
Electrodynamics General Circulation Model at 84–88 km (Liu et al., 2008, Fig. 5 and 6).



The temperature (Fig. 2b) shows a clear annual variation from the middle to the high
latitudes, with a minimum ~150 K at middle latitudes in July. The summer minimum at the
middle latitudes is an echo of those at high latitudes. The atomic oxygen concentrations (Fig.
2c) reveal the annual cycle. The concentrations have a maximum in winter and a minimum in
summer at high and middle latitudes, as has already been observed (Smith et al., 2010).
However, in the region from ~30° to ~50° N in summer, atomic oxygen concentrations show
one additional peak in June–July. Formation of this summer peak can be explained by the
transformed Eulerian mean (TEM) circulation (Limpasuvan et al., 2012, Fig. 7; Limpasuvan
et al., 2016, Fig. 5), which brings into the summer hemisphere the air reached by atomic
oxygen from the region of its production at high latitudes above 100 km to ~90 km at ~30°–
50° N. The peak altitude of the $OH^*_{v=6}$ (Fig. 2d) shows complex annual variability. There is a
secondary maximum OH* peak at ~30°–50° N in summer.
In order to assess the input into annual variability from different sources, we calculate relative
to annual averaged variations of volume emissions due to atomic oxygen, temperature, and air
density (Eq. A6). The derivation of these parameters is presented in the appendix. A similar
approach can be useful for analysing emission variations due to GWs, PWs, and tides.
Figure 3a shows relative variations of emissions due to impacts of atomic oxygen (black line),
temperature (red line), and air density (green line) at 43.75° N. The strongest emission
variation occurs because of changes in atomic oxygen concentration: the amplitude of its
relative deviation amounts to ~50%. The amplitudes of relative deviations of emissions due to
temperature and air density amount to ~15% and ~20%, respectively. The atomic oxygen
variation gives the most essential input into the winter maximum of emission (black line).
Because of the downward transport of atomic oxygen in winter, the volume emission rises by
~50 % averaged annually. The summer maximum is determined by the superposition of all
three factors. After the spring reduction of emissions due to the decline of atomic oxygen
concentration (~-40% of annual averaged values), the emissions rise again to approximately





the annual average values in June–July. This is synchronised with the growth of volume
emissions by ~20% over the annual average values due to summer temperature declines (red
line) and with the growth of volume emissions by ~15% over the annual average due to the
decline of peak altitude in April–September and the corresponding rise of air density (green
line).
Figure 3b illustrates relative variations of emissions due to second momenta (Eq. A7 in the
Appendix). The second momenta do not provide essential input to annual variation. The
strongest among them, $\frac{[O]'M'}{[O]\bar{M}}$ (blue line), gives emission variability with an amplitude ~6% of
annual averaged values.
In the context of our short paper, the ultimate question regarding the role of tides and GWs on
semi-annual variations of OH* emissions at middle latitudes has not been answered.
Undoubtedly, the simultaneous analysis of observations of excited hydroxyl emissions from
several stations is desirable to explore this question.

**4. Summary and conclusions**

Based on observations and numerical simulation, we confirmed the existence of a
semi-annual cycle of excited hydroxyl emission at middle latitudes with maxima in summer
(June–July) and winter (December–January). The annual variation in general mean circulation
and atomic oxygen concentration corresponding to the excited hydroxyl emission cycle was
found to be the leading cause of the winter maximum of this cycle, whereas the summer
maximum represents the superposition of three different processes: atomic oxygen meridional
transport due to residual circulation from the summer pole to the equator; temperature decline,
which represents the rest of the mesopause cooling at summer high latitudes; and air





concentration growth at the peak of the excited hydroxyl emission layer due to hydroxyl layer
descent at middle latitudes in April–September.

**Appendix.**

To obtain the derivation, we start with a simplified equation for excited hydroxyl
concentration. Taking into account that the ozone is in photochemical equilibrium in the
vicinity of the $[OH_v]$ layer and above during night-time (Kulikov et al., 2018; Belikovich et
al., 2018; Kulikov et al., 2019); utilising the equation for ozone balance during night-time
$(a_5[O_3][O] + a_1[H][O_3] = a_4[O][O_2][M])$, where $a_4$ and $a_5$ are the coefficients for the
corresponding reactions; omitting the reaction of atomic oxygen with ozone as relatively slow
(Smith et al., 2008); substituting the reduced ozone balance equation for the excited hydroxyl
balance equation (first term in the numerator of Eq. (1)); assuming that the most effective
production of excited hydroxyl occurs due to the reaction of ozone with atomic hydrogen and
that the most effective losses are due to quenching with molecular oxygen; we obtain from
Eq. (1) a simplified expression in which excited hydroxyl concentration is represented in
terms of atomic oxygen concentration, temperature (in $a_4$), and concentration of the
surrounding air:
$$[OH_v] \approx \mu_v a_4[O][M]. \tag{A1}$$
Here $\mu_v = \frac{\varsigma_v + \sum_{v'=v+1}^{v'=9} \mu_{v'} B_{v'v}}{\sum_{v''=0}^{v''=v-1} B_{vv''}}$, $(\varsigma_{v>9} = 0)$ are the coefficients representing the arithmetic
combination of branching ratios $\varsigma_v$ and quenching coefficients $B_{v'v}$. More comprehensive
derivations of (A1) can be found in a number of papers (Grygalashvyly et al., 2014;
Grygalashvyly, 2015; Grygalashvyly and Sonnemann, 2020). Although this is too simplified
to be used for precise calculations, it is useful for obtaining information about impacts and for
assessing variabilities.



By multiplying (A1) by the Einstein-coefficient $E_{vv''}$ for given a transition, writing the
reaction rate explicitly $a_4 = 6 \cdot 10^{-34}(300/T)^{2.4}$ (Burkholder et al., 2015), and collecting all
constants in $\chi_{vv''}$, we obtain an expression for volume emission in terms of atomic oxygen
concentration, temperature, and air number density:
$V \approx \chi_{vv''}T^{-2.4}[O][M],$ $(A2)$
where $\chi_{vv''} = \mu_v E_{vv''} \cdot 6 \cdot 10^{-34} \cdot 300^{2.4}$.
Next, we apply Reynolds decomposition by averaged and variable part to the temperature,
atomic oxygen concentration, and concentration of air in (A2):
$V \approx \chi_{vv''}(\bar{T} + T')^{-2.4}(\overline{[O]} + [O]')(\overline{[M]} + [M]'),$ $(A3)$
where $\bar{T}$, $\overline{[O]}$, $\overline{[M]}$ are average parts, and $T'$, $[O]'$, $[M]'$ are the corresponding varying parts.
After decomposing the term with temperature in the Taylor expansion and cross-multiplying
all terms of (A3), we obtain:
$V \approx \chi_{vv''}\bar{T}^{-2.4}\overline{[O]} \cdot \overline{[M]} + \chi_{vv''}\bar{T}^{-2.4}\overline{[O]}[M]' + \chi_{vv''}\bar{T}^{-2.4}[O]'\overline{[M]} - 2.4\chi_{vv''}T'\bar{T}^{-3.4}\overline{[O]} \cdot$
$\overline{[M]} + \chi_{vv''}\bar{T}^{-2.4}[O]'[M]' - 2.4\chi_{vv''}T'\bar{T}^{-3.4}\overline{[O]}[M]' - 2.4\chi_{vv''}T'\bar{T}^{-3.4}[O]'\overline{[M]} -$
$2.4\chi_{vv''}T'\bar{T}^{-3.4}[O]'[M]'.$ $(A4)$
The volume emission for a given transition can be represented as follows:
$V \approx \bar{V} + V'_M + V'_O + V'_T + V''_{OM} + V''_{TM} + V''_{TO} + higher\ momenta,$ $(A5)$
where, $\bar{V} = \chi_{vv''}\bar{T}^{-2.4}\overline{[O]} \cdot \overline{[M]}, V'_M = \chi_{vv''}\bar{T}^{-2.4}\overline{[O]}[M]', V'_O = \chi_{vv''}\bar{T}^{-2.4}[O]'\overline{[M]}, V'_T =$
$-2.4\chi_{vv''}T'\bar{T}^{-3.4}\overline{[O]} \cdot \overline{[M]}, V''_{OM} = \chi_{vv''}\bar{T}^{-2.4}[O]'[M]', V''_{TM} =$
$-2.4\chi_{vv''}T'\bar{T}^{-3.4}\overline{[O]}[M]', V''_{TO} = -2.4\chi_{vv''}T'\bar{T}^{-3.4}[O]'\overline{[M]}.$
Hence, relative deviations (RD) of emissions due to variations in atomic oxygen, temperature,
and concentration of air are:



$$RD'_O = 100\% \cdot \frac{V'_O}{\bar{V}} = 100\% \cdot \frac{[O]'}{\overline{[O]}},$$

$$RD'_T = 100\% \cdot \frac{V'_T}{\bar{V}} = 100\% \cdot -2.4\frac{T'}{\bar{T}}, \qquad (A6)$$

$$RD'_M = 100\% \cdot \frac{V'_M}{\bar{V}} = 100\% \cdot \frac{[M]'}{\overline{[M]}}.$$

The relative deviations (RD) of emissions due to second momenta are

$$RD''_{OM} = 100\% \cdot \frac{V''_{OM}}{\bar{V}} = 100\% \cdot \frac{[O]'[M]'}{\overline{[O][M]}},$$

$$RD''_{TM} = 100\% \cdot \frac{V''_{TM}}{\bar{V}} = 100\% \cdot -2.4\frac{T'[M]'}{\bar{T}\overline{[M]}}, \qquad (A7)$$

$$RD''_{TO} = 100\% \cdot \frac{V''_{TO}}{\bar{V}} = 100\% \cdot -2.4\frac{T'[O]'}{\bar{T}\overline{[O]}}.$$


**Data availability.** The data utilized in this manuscript can be downloaded from
http://ra.rshu.ru/files/Grygalashvyly_et_al_ANGEO_2020.
**Author contributions.** All authors contributed equally to this paper.
**Competing interests.** The authors declare that they have no conflict of interest.
**Acknowledgements.** This work was supported by the Russian Science Foundation (grant
#20-77-10006). Some data processing have been done under the state task of the Ministry of
science and higher education of the Russian Federation (project FSZU-2020-0009)".

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





**Figures**
Figure 1. Observed at 43° N (black line) and modelled at 43.75° N (red line), annual
variability of intensity and volume emission (a), temperature (b), atomic oxygen
concentration (c), and height at the peak of the OH$^*_{v=6}$ layer.

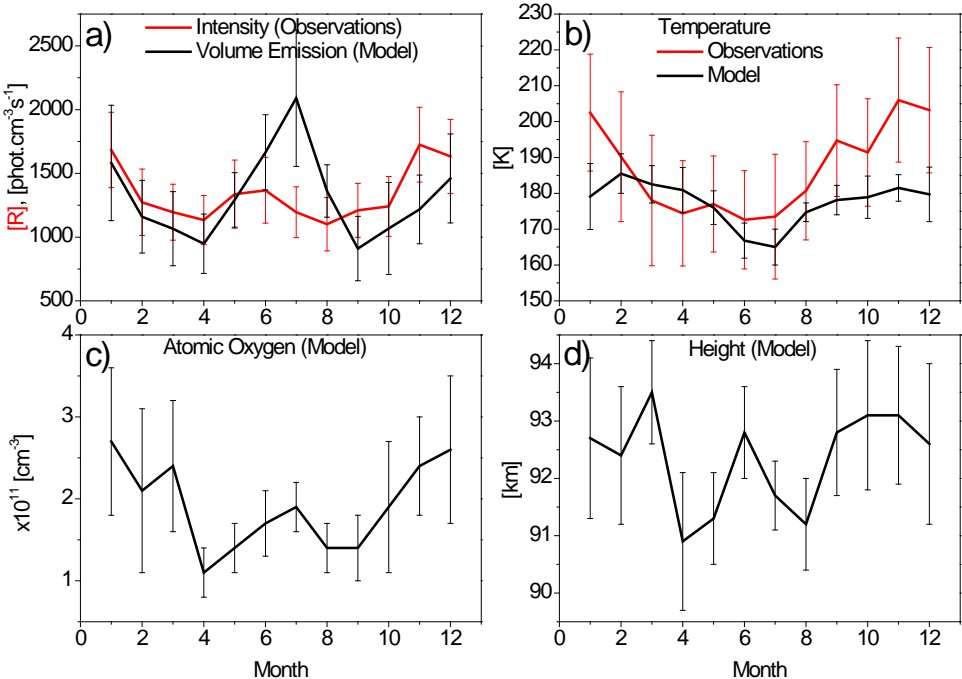
















Figure 2. Nightly mean one-month sliding average volume emission (a), temperature (b),
atomic oxygen at peak of OH$^*_{v=6}$ (c), and height of peak of OH$^*_{v=6}$.

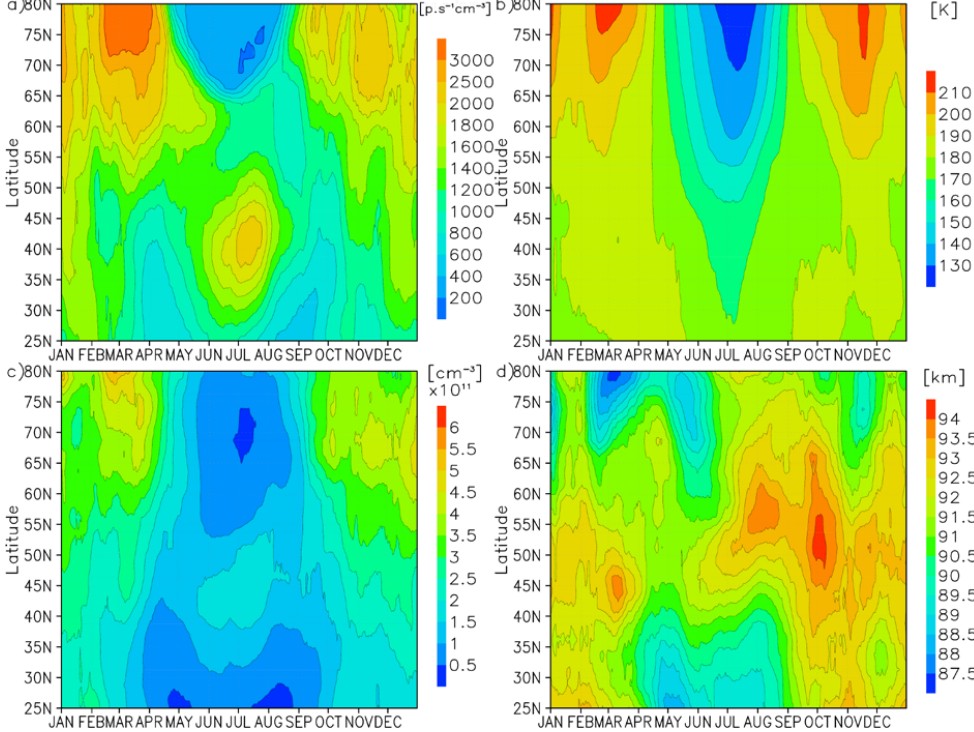

















Figure 3. a) relative to annual averaged variations of volume emission due to atomic oxygen
(black line), temperature (red line), and height (green line) at 43.75° N, b) relative variations
of volume emissions due to second momentum $\frac{[O]'M'}{[O]\bar{M}}$ (blue line), $\frac{T'M'}{\bar{T}\bar{M}}$ (cyan line), and $\frac{[O]'T'}{[O]\bar{T}}$
(magenta line) at 43.75° N.

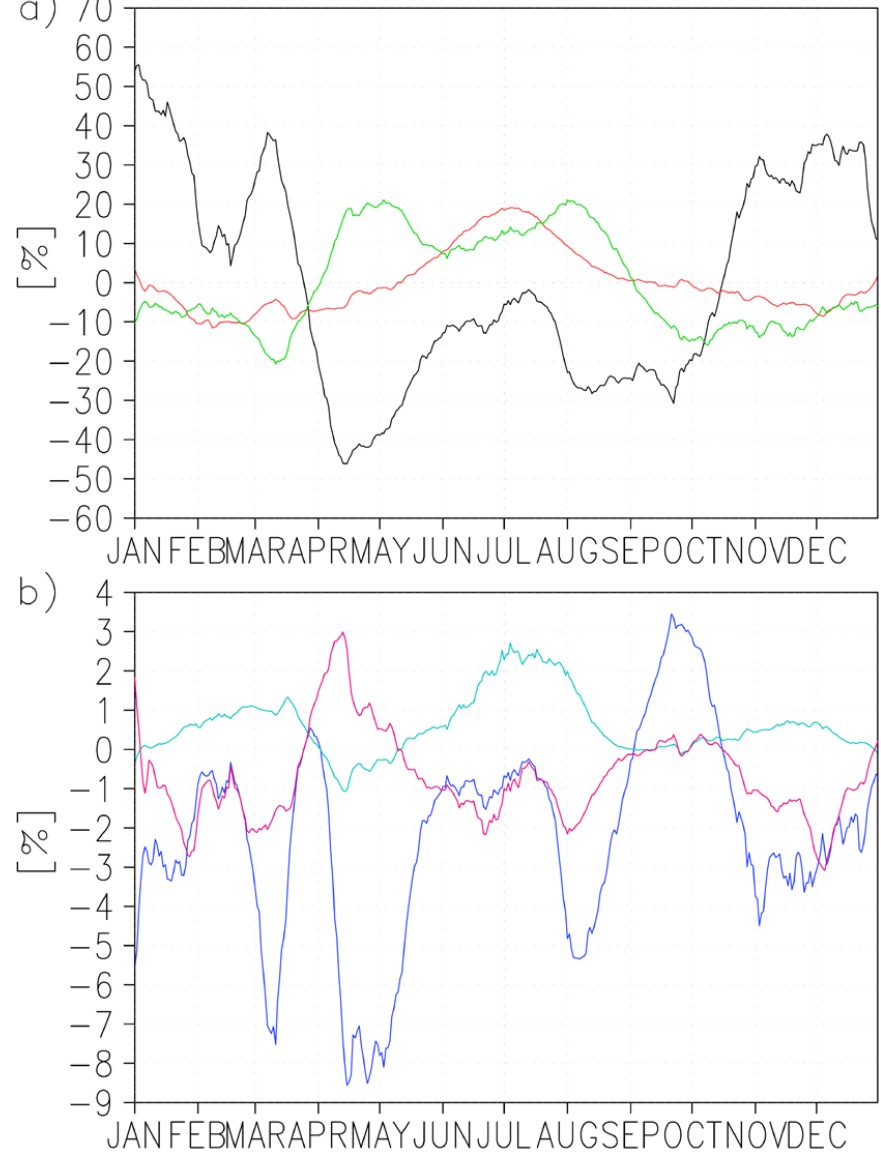
