# Peer review of "Semi-Annual Variation of Excited Hydroxyl Emission at Mid-Latitudes"

_Annales Geophysicae, 2020_

## Referee Comment (RC1) · Anonymous Referee #1 · 1 Jan 2021

In this paper, the annual dynamics of OH* emission at middle latitudes is analyzed, focusing on the causes of semi-annual cycle with maxima in summer and winter which have been revealed during previous long-term ground-based observations. First, the authors found a similar behavior of OH* emission in simulation data which were obtained with the use of the well tested 3-D chemical transport model of the middle atmosphere. Second, they developed a mathematical approach to analyze the causes driving the temporal evolution of OH* emission. Third, they found with the use of this approach that the observed features of the semi-annual cycle of OH* emission were caused mainly by the superposition of annual variations of temperature and O concentration at middle latitudes. The results obtained are interesting and therefore the paper merits publication. I have minor remarks only. 1. The authors analyze the simulation

data corresponding to year 2009. In what year or years were the experimental data shown in Figures 1 measured? 2. Tick labels (months) of all panels on Figs 2-3 have merged and are difficult to distinguish.

---

## Referee Comment (RC2) · Anonymous Referee #2 · 7 Jan 2021

Review of "Semi-Annual Variation of Excited Hydroxyl Emission at Mid-Latitudes" by Mykhaylo Grygalashvyly , Alexander I. Pogoreltsev, Alexey B. Andreyev, Sergei P. Smyshlyaev, and Gerd R. Sonnemann

The article presents experimental measurements of the airglow that corresponds to $OH\nu=6^*\rightarrow OH\nu=2^*$ transitions. The measurements taken at mid latitudes show semi-annual variation of the airglow. The authors reproduce this behavior in chemical model of MLT region. Using the model data the authors explain the behavior moreover they provide quantitative estimation of certain factors driving the airglow intensity.

The results presented are scientifically useful especially the quantitative estimations.

The article overall is solid but I have to point out that the presentation of the material

lucks some necessary details. The issues of the paper are seem to be caused by the desire of the authors to make the article shorter. The authors choose to provide numerous references instead of the detailed description. Which is of course acceptable. However some details cannot be deduced from the references so some clarification is still needed.

In my opinion the article can be published in Annales Geophysicae (ANGEO) after minor revision.

Here is the list of my complaints:

1) Experimental data set presented in the article is not fully described. Even the year of measurements is not mentioned anywhere in the article. Please add the description.

2) Part 2.2 should explain how volume emission rate is calculated from [OH$\nu$]. The reader should be devoid of guessing.

3) I believe the dynamic fields used as input data for the CTM should be explicitly characterized at least by reference.

4) Please comment on how the choice of the year in the run may affect the results.

5) The relation between (modeled) volume emission rate at peak and measured intensity should be explicitly described.

6) Please state the purpose of standard deviations on figure 1. They might deceive the reader as they look as measurement error.

7) I respect authors decision to move the formulas into appendix. (Though personally I can't agree with it as in my opinion they constitute the important part of the article.) However the A6 and A7 formulas should be placed in part 3 of the article and the notation ([O]', T', [M]') should be explained there. I also suggest the according modification of figure text (lines 628-631). Negative impact of certain factor is confusing without this.

Technical comments:

[Figure]

Line 86. I believe authors should mention emission rate retrieval here as well

Part 2.2 I suggest putting the description of the coefficients in (1) to the table for better readability.

Line 600. "Red" and "black" are clearly swapped here.

Figures 2 and 3. The time scale is hard to read (especially in figure 3) due to absence of space between month abbreviations.

Line 83 the representation of excited hydroxyl transitions differs from the other representations of the transition in the paper

I have to point out that I am not a native speaker so take the following suggestions with the grain of salt.

Line 167 Please add "of the peak" at the end. Line 171 "it represents the rest of the annual variation " I suggest "it represents the part of the annual variation" Line 179 "The summer minimum at the middle latitudes is the echo of the one at high latitudes" Line 200 May be "50% of annual average" Line 232 It is not clear at that point of the article the derivation of what is to be presented. I suggest rephrasing. Line 249-250 I suggest something like "Although the accuracy of (A1) estimate is insufficient for model calculations" as simplicity of the formula in general has nothing to do with the accuracy.

---

## Author Comment (AC1) · 22 Jan 2021

Response to the comments on the paper by Referee 1

Semi-Annual Variation of Excited Hydroxyl Emission at Mid-Latitudes By Mykhaylo Grygalashvyly, Alexander I. Pogoreltsev, Alexey B., Andreyev, Sergei P. Smyshlyaev, and Gerd R. Sonnemann

We appreciate the reviewer's constructive comments and positive judgment on our paper. We have taken the reviewer's suggestions into account when preparing the revised version of our manuscript.

In the following we address the comments of the reviewer point by point.

1. The data of observations represent nightly mean values averaged over years 2010-2017. We add such notation at line 93 of the revised manuscript. 2. Following by your suggestion the labels (months) of Figures 2 and 3 were corrected.

Other changes are related to the recommendations and demands of other referee. Thank you for taking the time to review our manuscript.

With respect, Mykhaylo Grygalashvyly, Alexander Pogoreltsev, Alexey Andreyev, Sergei Smyshlyaev, and Gerd Reinhold Sonnemann

Please also note the supplement to this comment:
https://angeo.copernicus.org/preprints/angeo-2020-80/angeo-2020-80-AC1-supplement.pdf

---

## Author Comment (AC2) · 22 Jan 2021

Response to the comments on the paper by Referee 2 Semi-Annual Variation of Excited Hydroxyl Emission at Mid-Latitudes By Mykhaylo Grygalashvyly, Alexander I. Pogoreltsev, Alexey B., Andreyev, Sergei P. Smyshlyaev, and Gerd R. Sonnemann

Dear Referee,

We appreciate positive judgment on our paper, constructive comments, and not formal approach to the review. We have taken your suggestions into account when preparing the revised version of our manuscript. In following we mention point by point how the manuscript has been changed according to your suggestions.

1. We add it at line 93 of the revised manuscript.

2. We add the explanation at lines 113-115 of the revised manuscript.

3. We add such description at lines 136-141 of the revised manuscript, as well necessary references in the reference list.

4. We add such a comment at lines 133-135.

5. We add such notation at lines 159-162.

6. We add such statements ant lines 156-157.

7. Following by your suggestion we add Eq. (A6) into the Section 3 with explanations about mean states and perturbations, as well we modified the description of the Fig. 3. We did not add the equation (A7) because second momentum have not essential impact on volume emission variability and in future investigations their consideration could be omitted.

Technical comments:

Line 86. This technical but very large problem was comprehensively described in large number of works of Lopez-Gonzalez, which we refer in our reference list.

Part 2.2. Following by your suggestion, we collected description of coefficients for Eq. (1) in the Table (1).

Line 600. Thank you for this note, it is true. We corrected the description of the Fig. 1.

Figures 2 and 3. We changed the time scale of these figures according with your suggestion.

Line 83. We change this nomenclature according with common nomenclature of our manuscript.

All of your language and stile corrections at lines 167, 171, 179, 200, 232, and 249-250 were applied completely.

Other changes are related to the recommendations and demands of other referee.

Thank you for taking the time to review our manuscript.

With respect, Mykhaylo Grygalashvyly, Alexander Pogoreltsev, Alexey Andreyev, Sergei Smyshlyaev, and Gerd Reinhold Sonnemann

Please also note the supplement to this comment: https://angeo.copernicus.org/preprints/angeo-2020-80/angeo-2020-80-AC2-supplement.pdf

---

## Author Comment (AC4) · 22 Jan 2021

Response to the comments on the paper by Referee 2 Semi-Annual Variation of Excited Hydroxyl Emission at Mid-Latitudes By Mykhaylo Grygalashvyly, Alexander I. Pogoreltsev, Alexey B., Andreyev, Sergei P. Smyshlyaev, and Gerd R. Sonnemann

Dear Referee,

We appreciate positive judgment on our paper, constructive comments, and not formal approach to the review. We have taken your suggestions into account when preparing the revised version of our manuscript. In following we mention point by point how the manuscript has been changed according to your suggestions.

1. We add it at line 93 of the revised manuscript: "The analysis presented in this paper

uses data averaged over the years 2010-2017."

2. We add the explanation at lines 113-115 of the revised manuscript: "We calculates volume emission for transition OH*v=6→OH*v=2 as the product of the Einstein coefficient for given transition by concentration of excited hydroxyl at corresponding vibrational number, i.e. ."

3. We add such description at lines 136-141 of the revised manuscript, as well necessary references in the reference list:" This run is based on the dynamics and temperature of LIMA (Leibniz Institute Middle Atmosphere) model for the so-called "realistic case", in which carbon dioxide, ozone, and Lyman-$\alpha$ flux are taken from observations, and the horizontal winds and temperature of ECMWF (European Centre for Medium-Range Weather Forecasts) are assimilated below ∼35 km (Berger, 2008; Lübken et al., 2009, 2013)."

4. We add such a comment at lines 133-135:" (the choice of this year does not affect our conclusions because calculations for other years show similar semi-annual variations)".

5. We add such notation at lines 159-162: "Note, that the observed intensity is directly proportional to the vertical integral of the volume emissions; hence, they reveal similar variations and dependencies on surrounding conditions near the peak of the excited hydroxyl layer."

6. We add such statements ant lines 156-157: "because we display monthly mean values and standard deviations commonly exceed the errors of measurements".

7. Following by your suggestion we add Eq. (A6) into the Section 3 with explanations about mean states and perturbations, as well we modified the description of the Fig. 3:" In order to assess the input into annual variability from different sources, we calculate relative to annual averaged variations of volume emissions due to atomic oxygen, temperature, and air density (Eq. A6):

where overbar denotes annually averaged values and prime denotes difference of actual (modeled or observed) values from annually averaged (in our case this is difference between nightly mean one month sliding averaged values (Fig. 2) and nightly mean annually averaged values)." We did not add the equation (A7) because second momentum have not essential impact on volume emission variability and in future investigations their consideration could be omitted.

Technical comments:

Line 86. This technical but very large problem was comprehensively described in large number of works of Lopez-Gonzalez, which we refer in our reference list.

Part 2.2. Following by your suggestion, we collected description of coefficients for Eq. (1) in the Table (1) and add in the text at lines 116-118 of the revised mynuscript: "All reactions used in Eq. (1) and in appendix, together with corresponding reaction rates, branching ratios, quenching rates and spontaneous emission coefficients, besides those for multi-quantum processes, are collected in Table 1."

Table 1. List of reactions with corresponding reaction rates (for three-body reactions [cm6 molecule−2 s−1] and for two-body reactions [cm3 molecule−1 s−1]), branching ratios, quenching coefficients, and spontaneous emission coefficients (s-1) used in the paper.

Line 600. Thank you for this note, it is true. We corrected the description of the Fig. 1.

Figures 2 and 3. We changed the time scale of these figures according with your suggestion.

Line 83. We change this nomenclature according with common nomenclature of our manuscript.

All of your language and stile corrections at lines 167, 171, 179, 200, 232, and 249-250 were applied completely.

Other changes are related to the recommendations and demands of other referee. Thank you for taking the time to review our manuscript.

With respect, Mykhaylo Grygalashvyly, Alexander Pogoreltsev, Alexey Andreyev, Sergei Smyshlyaev, and Gerd Reinhold Sonnemann

Please also note the supplement to this comment:
https://angeo.copernicus.org/preprints/angeo-2020-80/angeo-2020-80-AC4-supplement.pdf
* * *
1    **Table 1**. List of reactions with corresponding reaction rates (for three-body reactions [cm$^6$

2    molecule$^{-2}$ s$^{-1}$] and for two-body reactions [cm$^3$ molecule$^{-1}$ s$^{-1}$]), branching ratios, quenching

3    coefficients, and spontaneous emission coefficients (s$^{-1}$) used in the paper.

| | Reaction | Coefficient/branching ratios | Reference |
|---|---|---|---|
| 1 | $H + O_3 \xrightarrow{\varsigma_v a_1} OH_{v=5,\dots,9} + O_2$ | $a_1 = 1.4 \cdot 10^{-10} exp\left(\frac{-470}{T}\right)$
 $\varsigma_{v=9,\dots,5}$
 $= 0.47, 0.34, 0.15, 0.03, 0.01$ | Burkholder et al. (2015), Adler-Golden (1997) |
| 2 | $O + HO_2 \xrightarrow{\psi_v a_2} OH_{v=5,\dots,9} + O_2$ | $a_2 = 3.0 \cdot 10^{-11} exp\left(\frac{200}{T}\right)$
 $\psi_{v=3,\dots,1} = 0.1, 0.13, 0.34$ | Burkholder et al. (2015), Kaye (1988), Takahashi and Batista (1981) |
| 3 | $O + OH_{v=1,\dots,9} \rightarrow O_2 + H$ | $a_3(v = 9,\dots,5) = (5.07, 4.52, 3.87, 3.93, 3.22, 3.68, 3.05, 3.19, 3.42) \cdot 10^{-11}$ | Varandas (2004), Caridade et al. (2013) |
| 4 | $O + O_2 + M \rightarrow O_3 + M$ | $a_4 = 6 \cdot 10^{-34}(300/T)^{2.4}$ | Burkholder et al. (2015) |
| 5 | $O + O_3 \rightarrow 2O_2$ | $a_5 = 8 \cdot 10^{-12} exp\left(\frac{-2060}{T}\right)$ | Burkholder et al. (2015) |
| 6 | $OH_v + O_2, O, N_2 \rightarrow OH_{v' < v} + O_2, O, N_2$ | $B_{vv'}, D_{vv'}, C_{vv'}$ | Adler-Golden (1997), Caridade et al. (2013), Makhlouf et al. (1995) |
| 7 | $OH_v \rightarrow OH_{v' < v} + h\nu$ | $E_{vv'}$ | Xu et al. (2012) |

**Fig. 1.** Table 1

[Figure]

**Fig. 2.** Figure 2

**Fig. 3.** Figure 3

---

## Author Response (AR1)

Response to the comments on the paper by Referee 1

**Semi-Annual Variation of Excited Hydroxyl Emission at Mid-Latitudes**

By Mykhaylo Grygalashvyly, Alexander I. Pogoreltsev, Alexey B., Andreyev, Sergei P.

Smyshlyaev, and Gerd R. Sonnemann

We appreciate the reviewer's constructive comments and positive judgment on our paper. We have taken the reviewer's suggestions into account when preparing the revised version of our manuscript.

In the following we address the comments of the reviewer point by point.

1.  The data of observations represent nightly mean values averaged over years 2010-2017. We add such notation at line 93 of the revised manuscript: "The analysis presented in this paper uses data averaged over the years 2010-2017."
2.  Following by your suggestion the labels (months) of Figures 2 and 3 were corrected.

Other changes are related to the recommendations and demands of other referee.

Thank you for taking the time to review our manuscript.

With respect,

Mykhaylo Grygalashvyly, Alexander Pogoreltsev, Alexey Andreyev, Sergei Smyshlyaev, and

Gerd Reinhold Sonnemann

 Response to the comments on the paper by Referee 2

**Semi-Annual Variation of Excited Hydroxyl Emission at Mid-Latitudes**

By Mykhaylo Grygalashvyly, Alexander I. Pogoreltsev, Alexey B., Andreyev, Sergei P.

Smyshlyaev, and Gerd R. Sonnemann

Dear Referee,

We appreciate positive judgment on our paper, constructive comments, and not formal approach to the review. We have taken your suggestions into account when preparing the revised version of our manuscript. In following we mention point by point how the manuscript has been changed according to your suggestions.

1. We add it at line 93 of the revised manuscript: "The analysis presented in this paper uses data averaged over the years 2010-2017."

2. We add the explanation at lines 113-115 of the revised manuscript: "We calculates volume emission for transition $OH^*_{v=6} \rightarrow OH^*_{v=2}$ as the product of the Einstein coefficient for given transition by concentration of excited hydroxyl at corresponding vibrational number, i.e. $V_{62} = E_{62}[OH^*_6]$."

3. We add such description at lines 136-141 of the revised manuscript, as well necessary references in the reference list:" This run is based on the dynamics and temperature of LIMA (Leibniz Institute Middle Atmosphere) model for the so-called "realistic case", in which carbon dioxide, ozone, and Lyman-α flux are taken from observations, and the horizontal winds and temperature of ECMWF (European Centre for Medium-Range Weather Forecasts) are assimilated below ~35 km (Berger, 2008; Lübken et al., 2009, 2013)."

4. We add such a comment at lines 133-135:" (the choice of this year does not affect our conclusions because calculations for other years show similar semi-annual variations)".

5. We add such notation at lines 159-162: "Note, that the observed intensity is directly proportional to the vertical integral of the volume emissions; hence, they reveal similar variations and dependencies on surrounding conditions near the peak of the excited hydroxyl layer."

6. We add such statements ant lines 156-157: "because we display monthly mean values and standard deviations commonly exceed the errors of measurements".

7. Following by your suggestion we add Eq. (A6) into the Section 3 with explanations about mean states and perturbations, as well we modified the description of the Fig. 3:" In order to assess the input into annual variability from different sources, we calculate relative to annual averaged variations of volume emissions due to atomic oxygen, temperature, and air density

:

$$
\begin{aligned}
RD'_O &= 100\% \cdot \frac{V'_O}{\overline{V}} = 100\% \cdot \frac{[O]'}{\overline{[O]}}, \\
RD'_T &= 100\% \cdot \frac{V'_T}{\overline{V}} = 100\% \cdot -2.4\frac{T'}{\overline{T}}, \\
RD'_M &= 100\% \cdot \frac{V'_M}{\overline{V}} = 100\% \cdot \frac{[M]'}{\overline{[M]}},
\end{aligned}
\tag{2}
$$

where overbar denotes annually averaged values and prime denotes difference of actual (modeled or observed) values from annually averaged (in our case this is difference between nightly mean one month sliding averaged values (Fig. 2) and nightly mean annually averaged values)."

We did not add the equation (A7) because second momentum have not essential impact on volume emission variability and in future investigations their consideration could be omitted.

Technical comments:

Line 86. This technical but very large problem was comprehensively described in large number of works of Lopez-Gonzalez, which we refer in our reference list.

Part 2.2. Following by your suggestion, we collected description of coefficients for Eq. (1) in the Table (1) and add in the text at lines 116-118 of the revised mynuscript: "All reactions used in Eq. (1) and in appendix, together with corresponding reaction rates, branching ratios, quenching rates and spontaneous emission coefficients, besides those for multi-quantum processes, are collected in Table 1."

**Table 1**. List of reactions with corresponding reaction rates (for three-body reactions [$cm^6$

$molecule^{-2}$ $s^{-1}$] and for two-body reactions [$cm^3$ $molecule^{-1}$ $s^{-1}$]), branching ratios, quenching coefficients, and spontaneous emission coefficients ($s^{-1}$) used in the paper.

| | Reaction | Coefficient/branching ratios | Reference |
|---|---|---|---|
| 1 | $H + O_3 \xrightarrow{\varsigma_v a_1} OH_{v=5,...,9} + O_2$ | $a_1 = 1.4 \cdot 10^{-10} exp\left(\frac{-470}{T}\right)$

 $\varsigma_{v=9,...,5}$
 $= 0.47, 0.34, 0.15, 0.03, 0.01$ | Burkholder et al. (2015), Adler-Golden (1997) |
| 2 | $O + HO_2 \xrightarrow{\psi_v a_2} OH_{v=5,...,9} + O_2$ | $a_2 = 3.0 \cdot 10^{-11} exp\left(\frac{200}{T}\right)$
 $\psi_{v=3,...,1} = 0.1, 0.13, 0.34$ | Burkholder et al. (2015), Kaye (1988), Takahashi and Batista (1981) |
| 3 | $O + OH_{v=1,..,9} \rightarrow O_2 + H$ | $a_3(v = 9, ...,5) = (5.07,$
 $4.52, 3.87, 3.93, 3.22, 3.68,$
 $3.05, 3.19, 3,42) \cdot 10^{-11}$ | Varandas (2004), Caridade et al. (2013) |
| 4 | $O + O_2 + M \rightarrow O_3 + M$ | $a_4 = 6 \cdot 10^{-34}(300/T)^{2.4}$ | Burkholder et al. (2015) |
| 5 | $O + O_3 \rightarrow 2O_2$ | $a_5 = 8 \cdot 10^{-12} exp\left(\frac{-2060}{T}\right)$ | Burkholder et al. (2015) |
| 6 | $OH_v + O_2, O, N_2$
 $\rightarrow OH_{v'<v} + O_2, O, N_2$ | $B_{vv'}, D_{vv'}, C_{vv'}$ | Adler-Golden (1997), Caridade et al. (2013), Makhlouf et al. (1995) |
| 7 | $OH_v \rightarrow OH_{v'<v} + hv$ | $E_{vv'}$ | Xu et al. (2012) |

Line 600. Thank you for this note, it is true. We corrected the description of the Fig. 1.

Figures 2 and 3. We changed the time scale of these figures according with your suggestion.

Line 83. We change this nomenclature according with common nomenclature of our manuscript.

All of your language and stile corrections at lines 167, 171, 179, 200, 232, and 249-250 were applied completely.

Other changes are related to the recommendations and demands of other referee.

Thank you for taking the time to review our manuscript.

With respect,

Mykhaylo Grygalashvyly, Alexander Pogoreltsev, Alexey Andreyev, Sergei Smyshlyaev, and

Gerd Reinhold Sonnemann

[revised manuscript text omitted]